

# RNA interference-mediated silencing of genes involved in the immune responses of the soybean pod borer *Leguminivora glycinivorella* (Lepidoptera: Olethreutidae)

Ruixue Ran[1], Tianyu Li[1], Xinxin Liu[1], Hejia Ni[2], Wenbin Li[1,3] and Fanli Meng[1,3]

[1] Key Laboratory of Soybean Biology in the Chinese Ministry of Education, Northeast Agricultural University, Harbin, Heilongjiang, China
[2] Colleges of Life Science, Northeast Agricultural University, Harbin, Heilongjiang, China
[3] Key Laboratory of Biology and Genetics & Breeding for Soybean in Northeast China, Ministry of Agriculture, Northeast Agricultural University, Harbin, Heilongjiang, China

## ABSTRACT

RNA interference (RNAi) technology may be useful for developing new crop protection strategies against the soybean pod borer (SPB; *Leguminivora glycinivorella*), which is a critical soybean pest in northeastern Asia. Immune-related genes have been recently identified as potential RNAi targets for controlling insects. However, little is known about these genes or mechanisms underlying their expression in the SPB. In this study, we completed a transcriptome-wide analysis of SPB immune-related genes. We identified 41 genes associated with SPB microbial recognition proteins, immune-related effectors or signalling molecules in immune response pathways (e.g., Toll and immune deficiency pathways). Eleven of these genes were selected for a double-stranded RNA artificial feeding assay. The down-regulated expression levels of *LgToll-5-1a* and *LgPGRP-LB2a* resulted in relatively high larval mortality rates and abnormal development. Our data represent a comprehensive genetic resource for immune-related SPB genes, and may contribute to the elucidation of the mechanism regulating innate immunity in Lepidoptera species. Furthermore, two immune-related SPB genes were identified as potential RNAi targets, which may be used in the development of RNAi-mediated SPB control methods.

## INTRODUCTION

*Leguminivora glycinivorella* (Mats.) obraztsov (soybean pod borer (SPB)) belongs to the order Lepidoptera and family Olethreutidae. The SPB is the major pest of soybean in northeastern Asia (*Zhao et al., 2008*; *Meng et al., 2017a*). The larvae use the immature beans as a food source until they reach maturity, resulting in soybean yield losses of up to 40% (*Meng et al., 2017b*). Insecticides have been used to control SPB infestations over

Corresponding author
Fanli Meng, mengfanli@neau.edu.cn

the past three decades. However, larvae within soybean pods that are under a closed canopy are often not exposed to the applied insecticides. Because of the lack of effective SPB-resistant germplasm, conventional breeding has not resulted in the production of new SPB-resistant cultivars. Therefore, the SPB remains a major pest and is responsible for substantial soybean yield losses (*Wang et al., 2014*; *Song et al., 2015*). Consequently, soybean breeders and growers are interested in developing new strategies for controlling SPB infestations, with RNA interference (RNAi) representing a promising option (*Khajuria et al., 2015*; *Fishilevich et al., 2016*).

RNAi involves the degradation of specific endogenous mRNAs by homologous double-stranded RNAs (dsRNAs) (*Fire et al., 1998*). Depending on the function of the targeted gene, RNAi can inhibit insect growth or result in death (*Joga et al., 2016*; *Christiaens, Swevers & Smagghe, 2014*). RNAi is conserved in nearly all eukaryotic organisms, and feeding insect pests dsRNA molecules may be useful in protecting agriculturally important crops (feeding RNAi or plant-mediated RNAi) (*Mao & Zeng, 2014*; *Ulrich et al., 2015*). However, microinjection of RNAi is particularly successful when targeting genes involved in immune responses (*Terenius et al., 2011*). The effectiveness of the RNAi technique for controlling pests depends on whether appropriate candidate genes are targeted because RNAi efficacy and the RNAi signal transmission vary among genes (*Huvenne & Smagghe, 2010*). While applying RNAi technology to control Lepidoptera insects (i.e., moths and butterflies) has been problematic (*Shukla et al., 2016*), the information available regarding its efficiency has recently increased.

Insects such as *Drosophila melanogaster* and *Bombyx mori* have a vigorous innate immune system with which they use to defend against microbial infections (*Myllymäki, Valanne & Rämet, 2014*; *Parsons & Foley, 2016*; *Yang et al., 2017*; *Chen & Lu, 2018*). Peptidoglycan recognition proteins (PGRPs) are important pattern recognition receptors that detect peptidoglycan (PGN) in the cell walls of gram-negative and gram-positive bacteria. PGRPs activate the Toll or immune deficiency (IMD)/JNK pathways or induce proteolytic cascades that generate antimicrobial peptides (*Gao et al., 2015*; *Chen et al., 2014*). Antimicrobial peptides are critical for defending against invading pathogens and for protecting insects against infections (*Gegner et al., 2018*). However, little is known about SPB immune-related genes or the associated immune responses.

To identify the immune-related genes of SPB, we generated SPB transcriptome datasets based on Illumina sequencing. These datasets were used to identify many genes associated with microbe recognition, immune-related signalling, and defence effectors. Furthermore, RNAi was applied to study the effects of silencing immune-related genes on first instar larvae. A feeding assay involving an artificial diet supplemented with dsRNA was used to identify candidate target genes for controlling the SPB by RNAi.

## MATERIALS & METHODS

### Insect rearing

*L. glycinivorella* eggs collected from a naturally infested soybean field at the experimental station of Northeast Agricultural University in Harbin, China were hatched at 26 °C. The

resulting larvae were reared on an artificial diet prepared in our laboratory (*Meng et al., 2017a*). Adult moths were fed a 5% honey solution, and were allowed to oviposit on young bean pods. The first instar larvae were selected and subjected to artificial diet feeding experiments.

## Illumina sequencing

The T3 dsSpbP0 (double-stranded SBP ribosomal protein P0 RNAi transgenic soybean line) and wild-type 'DN50' plants, provided by the Key Laboratory of Soybean Biology of the Chinese Education Ministry, Harbin, China, were grown in a greenhouse at 24 ± 1 °C with 60% relative humidity under a 16-h light/8-h dark cycle (*Meng et al., 2017b*). At the R5 soybean stage (fully developed pods), three replicates of 50 first-instar larvae were reared on soybean pods of DN50 and T3 dsSpbP0 plants. All larvae were collected after three d and used to construct a cDNA library. cDNA library preparation and Illumina sequencing were conducted by the Biomarker Technology Company (Beijing, China). Briefly, total RNA was extracted from six pooled larvae using TRIzol reagent (Invitrogen, Carlsbad, CA, USA). The first-strand cDNA was synthesised using random hexamer-primers from purified Poly (A) mRNA. Second-strand cDNA was synthesised using DNA polymerase I and RNaseH, and then purified using a QiaQuick PCR extraction kit (Qiagen, Hilden, Germany). cDNA fragments of a suitable length (300–500 bp) were obtained by agarose gel electrophoresis and amplified by PCR to construct the final cDNA libraries using NEB Next Ultra RNA Library Prep Kit for Illumina (NEB, Ipswich, MA, USA). The cDNA library was sequenced on the Illumina HiSeq 2000 system (Illumina, San Diego, CA, USA). The unigenes from six samples were combined to create the SPB unigene database (*Chen et al., 2014*; *Meng et al., 2017c*). All raw transcriptome data have been deposited in the NIH Short Read Archive (accession numbers SRR5985984, SRR5985985, SRR5985986, SRR5985987, SRR5985988, SRR5985989).

## Identification of immune-related genes

A list of immune-related genes was compiled based on the available relevant literature (Table S1), and homologous *B. mori, D. melanogaster, Danaus plexippus* and *Papilio polytes* genes in the GenBank database were identified (*Guan & Mariuzza, 2007*; *Xu et al., 2012*). The tBLASTn algorithm-based tool was used to complete sequence similarity searches of the SPB transcriptome database (*Boratyn et al., 2013*). The 41 SPB immune-related genes were submitted to NCBI GenBank and their Accession numbers are shown in Table S1.

## Phylogenetic and domain analyses

Amino acid sequences were aligned with the Multiple Alignment program clustal omega (https://www.ebi.ac.uk/Tools/msa/clustalo/), and the phylogenetic tree was constructed in MEGA 5 based on the neighbour-joining method with 1,000 bootstrap replicates (*Tamura et al., 2011*; *Sievers et al., 2011*). The architecture of the protein domains was analysed using the SMART program (http://smart.embl-heidelberg.de/).

## dsRNA synthesis

We synthesised dsRNAs using the T7 RiboMAX Express Large Scale RNA Production System (Promega, Madison, WI, USA). The T7 RNA polymerase promoter sequence

was added to each end of the DNA templates during PCR amplifications. The primers containing the T7 RNA polymerase promoter were designed using Primer-BLAST (https://www.ncbi.nlm.nih.gov/tools/primer-blast/) (Table S2). For the negative control, the green fluorescent protein (*GFP*) gene was amplified from the PCAMBIA1302 vector as a template for *GFP* dsRNA synthesis. The template DNA and single-stranded RNA were eliminated from the transcription reaction by DNase I and RNase A, respectively. The prepared dsRNAs were purified by a phenol/chloroform extraction followed by an ammonium acetate precipitation. The dsRNAs were ultimately suspended in ultrapure water and quantified using a Nano Drop 2000 spectrophotometer (Thermo Scientific, Waltham, MA, USA).

## SPB feeding bioassay

The first instar SPB larvae were fed an artificial diet containing dsRNA (10 μg/g) for specific target genes as described by *Meng et al. (2017a)*. Control larvae were treated with the same concentration of *GFP* dsRNA. The feeding bioassay was completed in triplicate with 50 larvae per treatment or control. Three biological replicates were used for each treatment. The larvae were reared for 15 d at 26 °C under a 16-h light/8-h dark cycle with 65% relative humidity. The dsRNA-supplemented artificial diet was refreshed every three d. Body weight, mortality, and phenotypic abnormalities were recorded every three d.

## Quantitative real-time PCR (qRT-PCR)

For every treatment, two surviving larvae were randomly collected at each time point from all of the biological replicates from zero to 15 d after larvae were fed the artificial diet containing the target gene's dsRNA, frozen in liquid nitrogen and stored at −80 °C. Total RNA was extracted from two pooled larvae using the RNApure Tissue Kit (DNase I) (CWBIO, Beijing, China). Additionally, primer sets were synthesised (Table S2). The extracted RNA samples were treated with DNase I (Invitrogen, Carlsbad, CA, USA) to remove any contaminating genomic DNA. They were then used as the template for first-strand cDNA synthesis with the TIANScript RT Kit (Tiangen, Beijing, China). The qRT-PCR analysis was completed using the SYBR Green kit (Bio-Rad, Hercules, CA, USA) and a Roche LightCycler® 480 real-time PCR system (Roche, Basel, Switzerland). Each reaction used cDNA corresponding to 50 ng of total RNA and each primer at a final concentration of 100 nm in 20 μl reaction. Controls included non-RT controls (50 ng total RNA without reverse transcription was used to detect genomic DNA contamination) and non-template controls (water template).The qRT-PCR conditions were as follows: 95 °C for 5 min; 40 cycles at 95 °C for 30 s, 60 °C for 15 s, and 72 °C for 45 s; 95 °C for 1 min and 55 °C for 1 min. At the end of each qRT-PCR experiment, a melt curve was generated to check for primer-dimer formation. The efficiencies of the qRT-PCR primer pairs were greater than 90% (Table S2). The qRT-PCR analysis contained three biological replicates, each having three technical replicates (*Meng et al., 2017c*). Each target gene's qRT-PCR products were sequenced to confirm their identities.

Relative expression levels were calculated by the following formula using LightCycler 480 Software v1.5.0 (Roche, Basel, Switzerland): $R = 2^{-(\Delta Ct\,sample - \Delta Ct\,calibrator)}$, where R represents the relative expression level, $\Delta$Ct sample is the average difference between

the Ct of the gene and that of SBPß-actin in the experimental sample (*Meng et al., 2017b*), and ΔCt calibrator is the average difference between the Ct of the gene and the e SBPß-actin in the calibrator. A representative sample was set as the calibrator (*Bustin et al., 2009*). The differences in Ct values between technical replicates was less than 0.5. The relative expression level of 0d for each target gene was set as a benchmark. The relative expression level of other time point for each target gene was used to compare with 0d value.

## Statistical analyses

All of the data in this study are presented as mean ± SE. Significant differences were determined by one-way analysis of variance followed by least significant difference tests for mean comparisons. The statistical analysis was performed with SAS 9.21 software (SAS Institute, Cary, NC, USA). *P*-values were corrected by Bonferroni.

## RESULTS

### Identification of *iLeguminivora glycinivorella* immune-related genes

Amino acid sequences encoded by *D. melanogaster*, *B. mori,* and *Manduca sexta* (Lepidoptera) immune-related genes were used to search SPB transcriptome sequences. The 41 identified putative SPB immune-related genes were functionally classified into three groups, microbial recognition, immune signalling, and immune effector molecules (Table S1).

### Microbial recognition molecules

The PGRPs recognize conserved molecular patterns present in pathogens, but absent in the host, including PGNs, which are essential cell wall components of almost all bacteria. The PGRPs are encoded by a highly conserved gene family in insects, and are generally classified into two types (short and long) (*Dziarski & Gupta, 2006*; *Yang et al., 2017*). We identified eight SPB PGRPs, four short and four long types, with similarities to *D. melanogaster* PGRP-SC, PGRP-SD, and PGRP-LB (Fig. 1). Five (LgPGRP-SC1a, LgPGRP-SC1b, LgPGRP-SD1b, LgPGRP-LB1, and LgPGRP-LB2b) of the eight identified PGRPs were predicted to be secreted proteins, based on the presence of putative signal peptides, that function as amidases. LgPGRP-LB lacks a putative signal peptide, but contains a transmembrane region and amidase domain, suggesting that it serves as a transmembrane PGN receptor. In contrast, LgPGRP-SD1a and LgPGRP-LB2a carry only the PGRP domain, implying that they are intracellular proteins (Table S1).

The gram-negative bacteria-binding proteins (GNBPs) and β-1,3-glucan recognition proteins (βGRPs) belong to a subfamily of pattern recognition receptors and have strong affinities for the lipopolysaccharide of gram-negative bacteria and the β-1,3-glucan of fungi, but not for the PGN of gram-positive bacteria. Of the three GNBPs produced by *D. melanogaster* (GNBP1, GNBP2, and GNBP3), GNBP1 interacts with PGRP-SA to form a hydrolytic complex that activates the Toll pathway in response to gram-positive bacteria, while GNBP3 is required for detecting fungi and activating the Toll pathway (*Hughes, 2012*; *Rao et al., 2017*). We identified one *GNBP* gene in the SPB transcriptome datasets. The neighbour-joining phylogenetic analysis indicated that *LgGNBP3* is an ortholog of
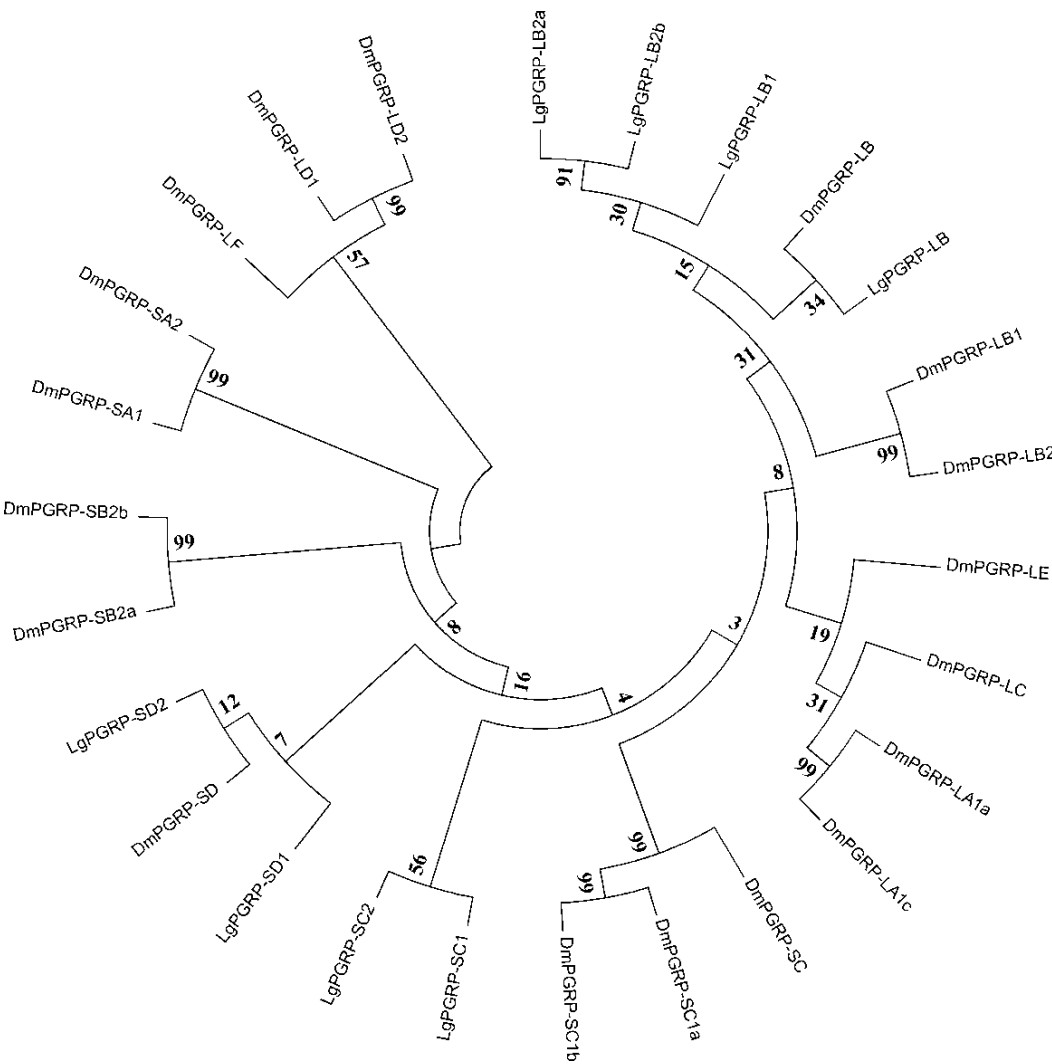

**Figure 1** **Phylogenetic relationships among PGRPs from *Leguminivora glycinivorella* and *Drosophila melanogaster*.** The phylogenetic tree was constructed using MEGA5.0 with a neighbour-joining approach. The bootstrap percentages (1,000 replicates) are provided next to the branches. The first two letters of each PGRP name indicate the species (Dm, *D. melanogaster*; Lg, *L. glycinivorella*).

*DmGNBP3* (Fig. S1). A comparison between the deduced amino acid sequences and the *D. melanogaster* GNBP sequences indicated that LgGNBP3 contains a putative N-terminal β-1,3-glucan-recognition domain (CBM39) and a C-terminal glucanase-like domain (glycosyl hydrolase family 16), suggesting that LgGNBP3 may bind to fungal β-1,3-glucan.

## Immune signalling molecules

After specific ligands are detected, microbial recognition molecules activate or modulate various immune response pathways. Here, we identified genes associated with the Toll and IMD pathways, which are the major signalling pathways that mediate the innate immunity of insects. The Toll pathway regulates the production of antimicrobial peptides in response

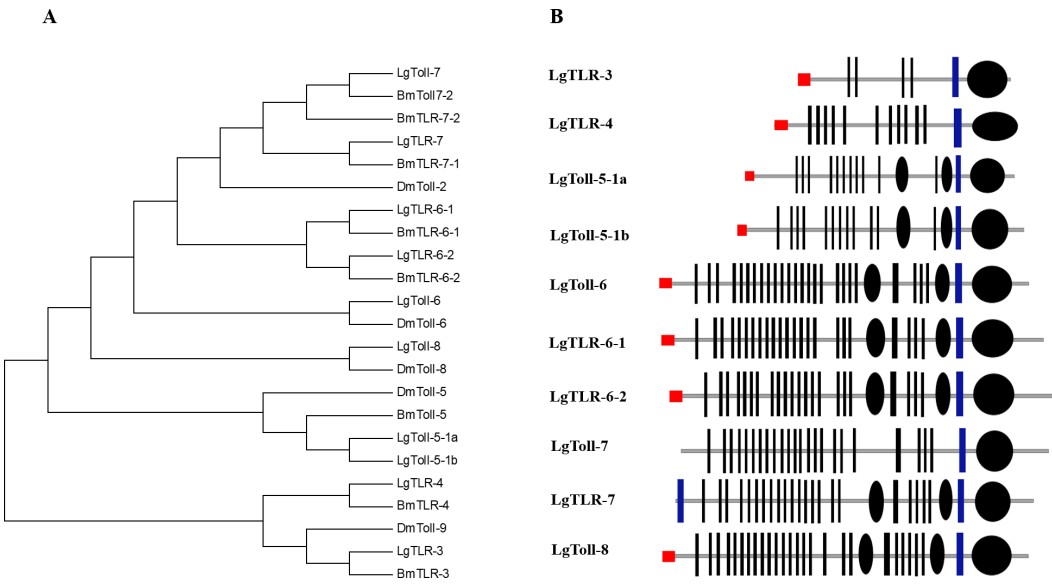

**Figure 2** **The analysis of Toll receptors of the *L. glycinivorella*.** (A) Phylogenetic relationships among Toll receptors from *Leguminivora glycinivorella*, *Bombyx mori* and *Drosophila melanogaster*. The phylogenetic tree was constructed using MEGA5.0 with a neighbour-joining approach. *Lg*, *L. glycinivorella*; *Bm*, *Bombyx mori*; *Dm*, *D. melanogaster*; TLR, Toll like receptor. (B) Predicted domains of the *L. glycinivorella* Toll receptors. The domain organization were predicted using the SMART program (http://smart.embl.de/). The extracellular leucine-rich repeats (LRRs), rectangles; LRR C-terminal domain, small ellipses; intracytoplasmic TIR domains, big ellipses; signal peptides, red rectangles; transmembrane domain, blue bar.

to infections by fungi or gram-positive bacteria with lysine-type PGNs in their cell walls (*Roh et al., 2009*). The Toll receptor, which is responsible for the signal transduction associated with the Toll pathway, is vital for insect innate immune responses and embryo development (*Takeda & Akira, 2004*; *Benton et al., 2016*). In this study, we identified 10 genes encoding Toll receptors in the SPB transcriptome datasets. The TIR domain is highly conserved in insect Toll families. To investigate the orthologous relationships among these genes, we constructed a phylogenetic tree based on an alignment of the TIR domains from all SPB and *D. melanogaster* Toll proteins. The Toll receptors analysed in this study formed six major clusters, namely Toll-3, Toll-4, Toll-5, Toll-6, Toll-7 and Toll-8 (Fig. 2A). Based on the phylogenetic tree, the *L. glycinivorella Toll* genes were designated as *LgTLR-3*, *LgTLR-4*, *LgToll-5-1a*, *LgToll-5-1b*, *LgToll-6*, *TLR-6-1*, *TLR-6-2*, *LgToll-7*, *LgTLR-7* and *LgToll-8*. All 10 predicted proteins contain an extracellular LRR domain as well as transmembrane and cytoplasmic TIR domains (Fig. 2B). We also identified sequences matching the intracellular components, ECSIT and Tollip, which affect the Toll signalling pathway (Table S1).

The IMD pathway is mainly activated by gram-negative bacterial infections. Additionally, IMD signal transduction is reportedly mediated by IMD, fas-associated death domain protein (FADD), death-related ced-3/Nedd2-like caspase (Dredd), inhibitor of apoptosis protein 2 (IAP2), transforming growth factor β-activated kinase (TAK1), TAK1-binding 2 (Tab2), ubiquitin conjugating 13 (Ubc13), and an inhibitor of nuclease factor B kinase

subunits b and g (IKKb and IKKg) (*Bao et al., 2013*; *Myllymäki, Valanne & Rämet, 2014*). Of these,we only identified sequences that were homologous to *FADD*, *Dredd*, and *IAP2* (Table S1).

### Immune-related effector genes

The PGRPs and βGRPs detect PGNs and β-1,3-glucans, which activate a clip-domain serine protease (CLIP) cascade that converts prophenoloxidase to active phenoloxidase, leading to the melanisation responses involved in eliminating pathogens (*Monwan, Amparyup & Tassanakajon, 2017*; *Li et al., 2016*). We identified two *CLIP* genes (*LgSnake-1* and *LgSnake-2*) in the SPB transcriptome datasets. The deduced amino acid sequences each contain a clip domain at the N-terminus and a serine protease domain at the C-terminus (Fig. S2). Serine protease inhibitors (i.e., serpins) negatively regulate prophenoloxidase activation, which prevents the excessive activation of the CLIP cascade. In this study, we identified three serpin genes, *Lgserpin1*, *Lgserpin2*, and *Lgserpin3*, in the SPB transcriptome datasets. Their deduced amino acid sequences each contained a putative signal peptide sequence and a core serpin domain, suggesting that they are secreted proteins (Table S1).

### Immune response effector genes

Antibacterial peptides are immune response effectors that are induced by immune challenges and are important for defence responses against insects. Diverse antibacterial peptide genes have been identified in many insect species, including genes encoding defensins, reeler, and lysozyme (*Imler & Bulet, 2005*; *Bao et al., 2011*). In this study, we identified two defensin genes, *Lgdefensin1* and *Lgdefensin2*, in the SPB transcriptome datasets. The encoded amino acid sequences each consisted of a putative signal peptide sequence and a core Knot1 domain (Table S1). We also identified seven chicken-type (C-type) lysozymes and two invertebrate-type (I-type) lysozymes in the *L* SPB transcriptome. The C-type lysozymes are bacteriolytic enzymes that hydrolyse the β (1–4) bonds between N-acetylglucosamine and N-acetylmuramic acid in the PGN of prokaryotic cell walls. The predicted SPB C-type proteins, with the exception of the C-type 3 protein, each include an N-terminal signal peptide sequence (Table S1). Additionally, we detected eight conserved cysteine residues in the *L. glycinivorella* C-type lysozymes (Fig. S3A) as well as 12 conserved cysteine residues in the deduced SPB I-type lysozyme sequences (Fig. S3B). These cysteine residues may form intramolecular disulphide bonds to enhance stability and resistance against proteolytic degradation.

### Potential RNAi targets identified in an artificial feeding assay and effects of double-stranded RNA on soybean pod borer development and mortality

In total, 11 genes representing the immune-related SPB genes were selected and analysed to identify potential new RNAi targets useful for controlling the SPB. We synthesised the corresponding dsRNAs in vitro and mixed them in an artificial diet. The mortality rates three d after larvae were fed artificial diets, which contained containing 10 μg/g dsRNA for *LgPGRP-LB*, *LgPGRP-LB2b*, *LgToll-5-1a*, *LgToll-5-1b*, *LgTLR-7*, *LgSerpin2* or *LgChaoptin* were 37%–92%. These mortality rates were significantly greater than those

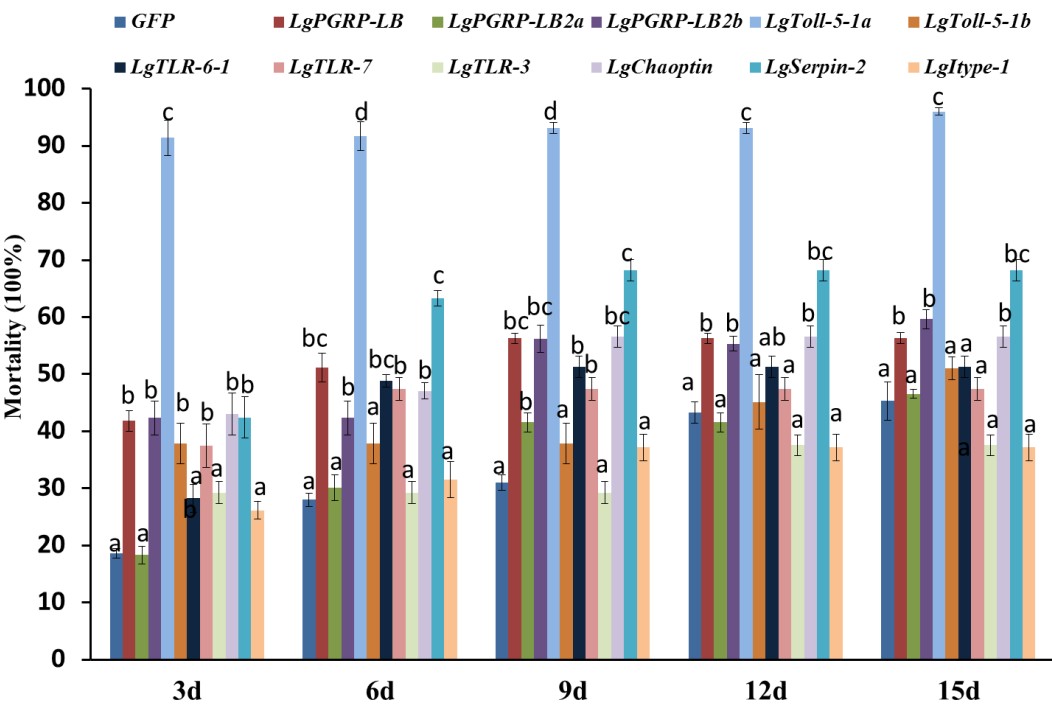

**Figure 3** **Mortality rates of soybean pod borer larvae fed an artificial diet supplemented separately with dsRNA (10 μg/g) for 11 candidate RNA interference target genes.** Columns represent mean ± SE. Different letters above the column indicate significant difference ($p < 0.0009$).

of control larvae treated with *GFP* dsRNA. Additionally, the final mortality rates of the larvae fed *LgPGRP-LB*, *LgPGRP-LB2b*, *LgToll-5-1a*, *LgSerpin2* and *LgChaoptin* dsRNA were even higher at 15 d. In contrast, the artificial diets containing dsRNA independently targeting *LgPGRP-LB2a*, *LgTLR-3* and *Lgitype-1* did not have any statistically significant effects on larval mortality (Fig. 3). Moreover, two main phenotypic differences were observed among the surviving larvae after 15 d of feeding. First, the weights of the larvae fed *LgToll-5-1b* and *LgItype-1* dsRNA increased more gradually than those of larvae fed *GFP* dsRNA, and was ultimately lower after 15 d of feeding (Fig. 4). Additionally, the *LgToll-5-1b* RNAi treatment resulted in partly black cuticles. Second, larvae fed dsRNA targeting *LgPGRP-LB* or *LgPGRP-LB2a* underwent early pupation, with pupation rates of 25 and 50%, respectively. The remaining larvae developed abnormally with stunted and twisted bodies (Fig. 5).

To investigate how larval mortality and abnormal development are correlated with the relative expression levels of specific target genes, we performed qRT-PCR using total RNA extracted from the surviving larvae at different time points after feeding on artificial diets. The expression levels of the genes, except for those of *LgPGRP-LB2a*, *LgTLR-6-1 and LgChaoptin* decreased in larvae three d after being treated with the respective dsRNAs, while the expression levels of all of the genes decreased significantly after six or nine d (Fig. 6). Thus, the increased mortality rates and abnormal development of larvae fed dsRNA may

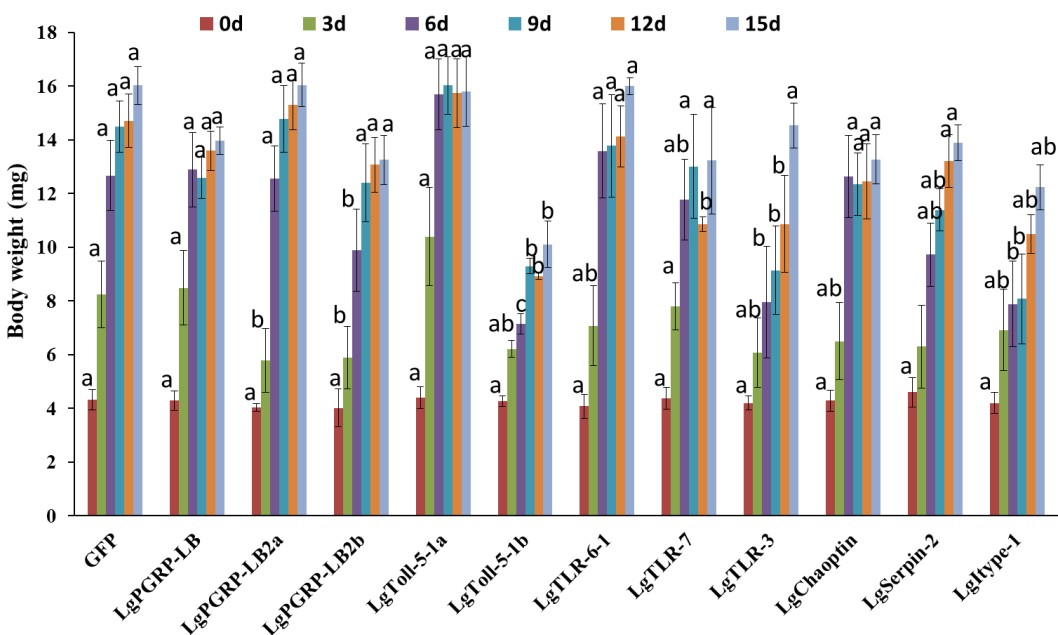

**Figure 4** **Body weights of soybean pod borer larvae fed artificial diets supplemented separately with dsRNA (10 µg/g) for 11 candidate RNA interference target genes.** Columns represent mean ± SE. Different letters above the column indicate significant difference ($p < 0.0009$).

result from the down-regulated expression of specific target genes. Moreover, unigenes *LgToll-5a* and *LgPGRP-LB2a* may represent good RNAi targets for controlling the SPB.

## DISCUSSION

Insects possess efficient innate immune systems that protect them from microorganisms and aid in abiotic stresses (*Hillyer, 2015*; *Parsons & Foley, 2016*). In this study, we identified 41 genes in the SPB transcriptome that encode components of conserved immune signalling pathways (Toll and IMD pathways) as well as pathogen recognition and immune response effectors. Most of these genes contained conserved sequences that exist in orthologous *D. melanogaster* and *B. mori* genes (Table S1). However, immune-related gene families have expanded or contracted in different taxa. For example, the *PGRP* gene families in *D. melanogaster*, *B. mori*, SPB have 13, 12, and eight members, respectively (*Hillyer, 2015*; *Yang et al., 2015*). In addition to sequence differences among the immune-related genes, the encoded proteins exhibited diverse activities. For example, four of the SPB's PGRPs are closely related to each other and form an independent cluster with *D. melanogaster* PGRP LB (Fig. 1). Two of them contain a putative signal peptide and a conserved Ami_2 domain, while the others lack a signal peptide (Table S1). Furthermore, silencing *LgPGRP-LB* and *LgPGRP-LB2a* induced early pupation and abnormal larval development, while silencing *LgPGRP-LB2b* lead to significantly higher mortality rates at three d and six d, indicating that *LgPGRP-LB2b* may be essential for early larval development (Fig. 5). PGRP-LB is a catalytic amidase that can degrade PGN and regulate host immune responses to infectious

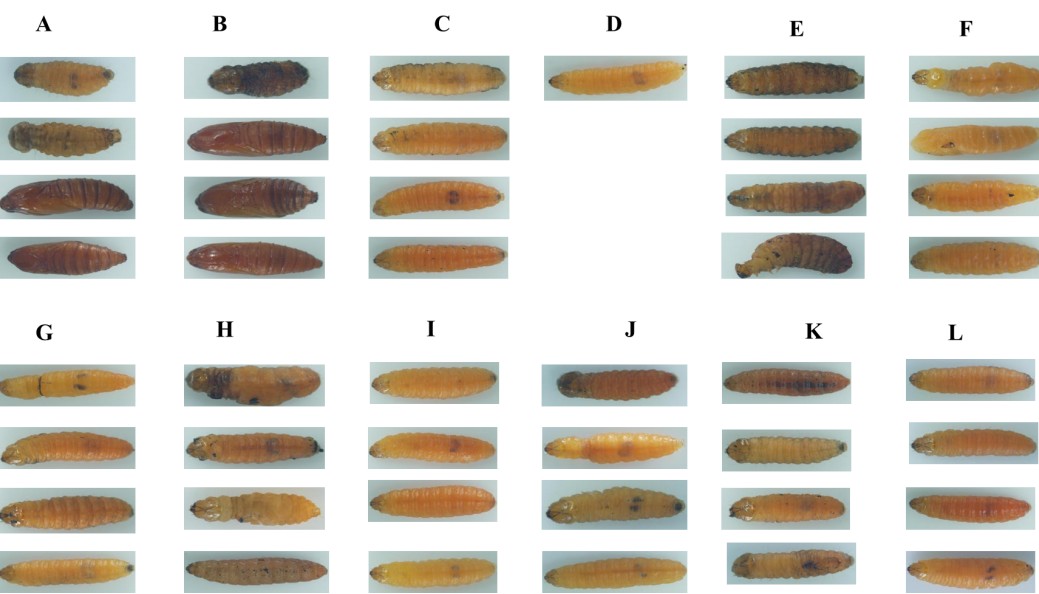

**Figure 5** **Images of soybean pod borer larvae fed an artificial diet supplemented separately with dsRNA (10 µg/g) for 11 candidate RNA interference target genes for 15 d.** (A) The larvae fed an artificial diet containing dsRNA for *LgPGRP-LB*. (B) The larvae fed an artificial diet containing dsRNA for *LgPGRP-LB2a*. (C) The larvae fed an artificial diet containing dsRNA for *LgPGRP-LB2b*. (D) The larvae fed an artificial diet containing dsRNA for *LgToll-5-1a*, only one survived after 15 d. (E) The larvae fed an artificial diet containing dsRNA for *LgToll-5-1b*. (F) The larvae fed an artificial diet containing dsRNA for *LgTLR-6-1b*. (G) The larvae fed an artificial diet containing dsRNA for *LgTLR-7*. (H) The larvae fed an artificial diet containing dsRNA for *LgTLR-3*. (I) The larvae fed an artificial diet containing dsRNA for *LgChaoptin*. (J) The larvae fed an artificial diet containing dsRNA for *LgSerpin-2*. (K) The larvae fed an artificial diet containing dsRNA for *LgItype-1*. (L) The larvae fed an artificial diet containing dsRNA for *GFP*.

microorganisms by down-regulating the IMD pathway (*Zaidman-Rémy et al., 2006*; *Troll et al., 2009*; *Gendrin et al., 2017*), which protects the beneficial microbes in insects and prevents host-inflicted damage during development (*Hashimoto et al., 2007*). In the Tsetse fly (Diptera: Glossinidae), silencing *PGRP-LB* by RNAi decreases host fecundity because of the associated cost of activating the host immune response (*Wang & Aksoy, 2012*).

The SPB is a univoltine insect. The mature larvae make cocoons in the soil, enter diapause during the winter and pupate in mid-July, resulting in a diapause period of 10 months (*Meng et al., 2017b*). In our study, *LgPGRP-LB* and *LgPGRP-LB2a* were silenced by RNAi, which broke diapause and caused mature larvae to pupate. This termination of diapause result from an immune response that was initiated to prevent host-inflicted damage. Further research is needed to confirm that LgPGRP-LB influences the host immune responses' activation.

The Toll pathway is critical for innate immunity against bacteria and also affects embryonic development, olfactory neuron processes, and TNF-induced JNK-dependent cell death in *D. melanogaster* (*Yang et al., 2015*; *Valanne, Wang & Rämet, 2011*; *Wu et al., 2015*). Knocking down the *fusilli* and *cactin* genes, which are part of the Toll pathway, is lethal for the red flour beetle (*Tribolium castaneum*), and the silencing *cactin* is 100% lethal

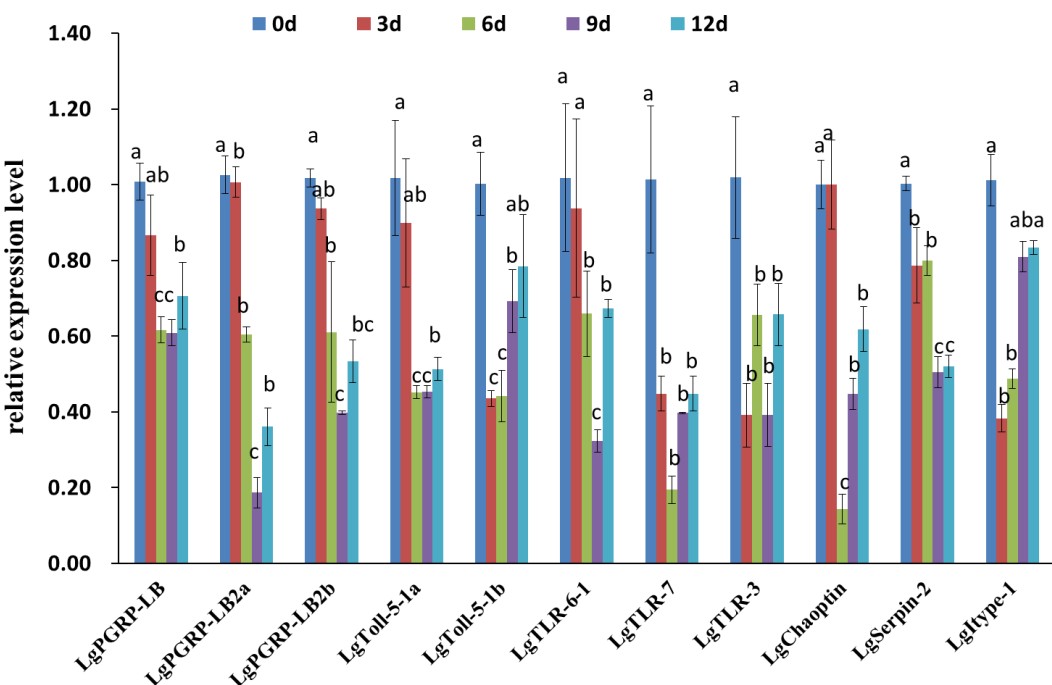

**Figure 6  Relative expression levels of 11 candidate RNA interference target genes at different time points after soybean pod borer larvae were fed separate artificial diets containing the respective dsRNA (10 μg/g).** Quantitative real-time polymerase chain reactions were completed using total RNA extracted from surviving larvae. Columns represent mean ± SE. Different letters above the column indicate significant difference ($p < 0.002$).

at all developmental stages (from larva to adult). Additionally, the knockdown of *pelle* and *dorsal* prevents eggs from hatching in the next generation (*Bingsohn et al., 2017*). In our study, a 10% knocked down of *LgToll-5-1a* lead to 92% larval mortality rate three d after feeding. While *LgToll-5-1a* is more critical for the first instar larvae development, a 56% knocked down of *LgToll-5-1b* did not impact on the survival rate of larvae three d after feeding. Knocking down *LgToll-5b* took considerably longer (15 d) to have an impact on body weights, and it prevented old cuticles from separating from larval bodies. *LgToll-5-1b* may influence mid-to-late larval development. Thus, *LgToll-5* may play critical roles in larval development, and it may function in immunity in adults. A future study will challenge *Toll-5-1a* or *LgToll-5-1b* RNAi-treated insects with pathogen infections to determine their roles, if any, in immunity.

Lysozymes are widely distributed immune effectors that exhibit muramidase activities against the PGNs in bacterial cell wall to induce cell lysis (*Zhou et al., 2017*). In our study, the LgI-type-1 gene encodes a destabilase domain, which is associated with isopeptidase and antibacterial activities. The pI of LgI-type-1 is 7.93 (Table S1). Researchers have proposed that I-type lysozymes with high pI values influence immunity (*Kurdyumov et al., 2015*; *Xue et al., 2004*). Thus, LgI-type-1 may have isopeptidase activity and play a role in the SPB's immune system.

## CONCLUSION

We identified 41 genes associated with SPB microbial recognition proteins, immune-related effectors, or signalling molecules of immune response pathways (e.g., Toll and immune deficiency pathways). This will be useful as a comprehensive genetic resource for immune-related SPB genes and may help elucidate the mechanism regulating innate immunity in Lepidoptera species. In addition, the in vivo functions of 11 genes were analysed in RNAi experiments, which indicated that three genes may be appropriate RNAi targets for controlling the SPB. The observations described herein may be useful for future analyses of the mechanisms underlying the SPB mmune response pathways and for developing RNAi-mediated methods to control SPB infestations.

## ACKNOWLEDGEMENTS

We thank Professor Xiaoyun Wang (Northeast Agricultural University, China) for her technical support in culturing the SPB.

### Funding

This work was financially supported by China National Novel Transgenic Organisms Breeding Project (2016ZX08004-004-006), the Chinese National Natural Science Foundation (31201229), the Foundation for University Key Teacher of the Department of Education of Heilongjiang Province (1253G008), the Academic Backbone Project of Northeast Agricultural University (15XG03), and Harbin Science and technology innovation program (2017RAQXJ122). The funders had no role in study design, data collection and analysis, decision to publish, or preparation of the manuscript.

### Grant Disclosures

The following grant information was disclosed by the authors:
China National Novel Transgenic Organisms Breeding Project: 2016ZX08004-004-006.
Chinese National Natural Science Foundation: 31201229.
Foundation for University Key Teacher of the Department of Education of Heilongjiang Province: 1253G008.
Academic Backbone Project of Northeast Agricultural University: 15XG03.
Harbin Science and technology innovation program: 2017RAQXJ122.

### Competing Interests

The authors declare there are no competing interests.

### Author Contributions

- Ruixue Ran performed the experiments, analyzed the data, prepared figures and/or tables, authored or reviewed drafts of the paper, approved the final draft, was the photographer of Figure 5 (images of larvae fed an artificial diet supplemented with dsRNA).

- Tianyu Li performed the experiments, prepared figures and/or tables, authored or reviewed drafts of the paper, approved the final draft.
- Xinxin Liu analyzed the data, prepared figures and/or tables, authored or reviewed drafts of the paper, approved the final draft.
- Hejia Ni performed the experiments, analyzed the data, prepared figures and/or tables, authored or reviewed drafts of the paper, approved the final draft.
- Wenbin Li contributed reagents/materials/analysis tools.
- Fanli Meng conceived and designed the experiments, contributed reagents/materials/analysis tools, prepared figures and/or tables, authored or reviewed drafts of the paper, approved the final draft.

### Data Availability

Sequences have been deposited at GenBank with accession numbers: SRR5985984, SRR5985985, SRR5985986, SRR5985987, SRR5985988, SRR5985989.

### Supplemental Information

Supplemental information for this article can be found online at http://dx.doi.org/10.7717/peerj.4931#supplemental-information.

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
