# Peer review of "RNA interference-mediated silencing of genes involved in the immune responses of the soybean pod borer Leguminivora glycinivorella (Lepidoptera: Olethreutidae)"

_PeerJ, doi:10.7717/peerj.4931_

## Round 0.1 · original submission · Minor Revisions

Thank you for submitting your manuscript to PeerJ. Based on the thorough comments of three reviewers and my own reading of your manuscript I invite you to resubmit after making minor revisions. Your paper meets the editorial guidelines for PeerJ and all three reviewers identified its strengths. Two of the three reviewers noted some major issues that should be addressed in a resubmission and all three detailed a number of minor issues. Please respond to all reviewer comments in your resubmission. The following points seem most important:

Reviewer 1 noted that no data are presented to show that knockdown of these immune–related genes increases susceptibility to pathogens. They recommend that without new experiments you make clear that these genes have not been shown to be involved in immunity in the soybean pod borer (SPB). Reviewer 1 has several other questions about data interpretation and presentation that should be addressed.

Reviewer 2 provided the most extensive comments. Some of them are minor points that will likely improve the final manuscript and should be addressed. But some points are central to the interpretation of your data. Please address this reviewer’s concerns about how statistics are described, conducted and presented in figures 3, 4 and 6. The reviewer raises a serious concern about how the p-values are adjusted to account for multiple hypotheses. They also ask that the type of error bar used be specified in the figure legends. The figure bars are not equal around each column, making it appear that these are not +/- bars. This detail needs to be clarified. Two of these three figure legend state that a student’s t-test was used, but the methods mentions an ANOVA. Please clarify exactly how these statistics were used and how they are presented. The reviewer notes that figure 6 seems to show expression levels of GFP, which does not seem to make sense. Please clarify this point as well and explain how fold differences in gene expression were calculated.

I can confirm Reviewer 2’s point that while four supplementary tables are referenced in the manuscript only two such tables seem to have been submitted. Please revise your table citations or upload the missing tables with your resubmission.

Reviewer 3 has asked that accession numbers for sequences submitted to a public database be added.

I have just a few suggestions of my own:

1. I would suggest that some detail be added about cDNA library construction to the Methods section. While reference is made to a previous study, it might help the reader to provide some basic description for library construction (kit used, approach, et.).

2. Could you add a citation showing that beta actin is a consistent endogenous control to use in qRT-PCR for the SPB? I would also describe whether and/or how tissue samples were stored between collection and RNA purification. Please also mention the enzyme used for qPCR, the machine used and the concentrations of cDNA and primer used. Were PCR products sequenced to confirm their identity? Were non-template controls and –RT controls used? Table S4 showing the efficiency of the qPCR primers was not included with the uploaded files.

3. It would be helpful to describe what software was used to calculate Ct values, delta Ct values and fold changes in expression. Were default settings used? What was considered an acceptable range in Ct values between technical replicates?

Reviewer 1 ·

Basic reporting

Manuscript passes on all criteria.

Experimental design

Pass

Validity of the findings

Pass - please see comments concerning some minor revisions to clarify significance of findings.

Additional comments

The manuscript by Ran et al. entitled “RNA interference-mediated silencing of genes involved in the immune responses of the soybean pod borer Leguminivora glycinivorella (Lepidoptera: Olethreutidae)” describes the identification of innate immunity genes from a transcriptomic analysis of a lepidopteran pest species, and provides evidence that RNAi-mediated knockdown of some of these immunity genes has serious impacts on the survival of the treated insects. The authors identified 41 immune-related genes using sequence comparisons and phylogenetic analyses with other known immune genes from other insects. The gene sequences’ identities were further supported by examination of predicted modular domains, providing additional evidence for homology with other known immune genes. The authors selected 11 genes to target for RNAi-mediated knockdown, and demonstrated that several genes were essential for development and growth, and that the dsRNAs could potentially be used for insecticidal control of this pest. Given that RNAi has been particularly challenging in many lepidopteran insects, their demonstration of RNAi efficacy in this insect, while previously documented in one of their previous studies, is still worthy of note, as this study illustrates the potency of immunity genes as possible RNAi targets for pest insect control.
The experiments were well described and conducted. The paper was very well written, with some good explanations for many of their findings. I believe that the paper is definitely worthy of publication, as it has provided an excellent application of transcriptomic analyses with its RNAi bioassays. I have only a few suggestions/questions for them to address:

Main points (page numbers refer to the PDF document, including PeerJ’s instructions pages)
1. On page 19 the authors state: “… larvae treated with Lgitype-1 dsRNA were more sensitive to pathogens, with approximately 50% of the larvae infected by bacteria even in the absence of an artificial pathogen inoculation.” There are no data provided in the manuscript to support this statement, and no mention in the Methods of how bacterial infections were observed or measured. There was no mention in the Methods of artificial pathogen inoculation. To provide definitive proof that the genes under examination in their study are involved in immunity, the best approach would be to challenge the RNAi-treated insects with infections and determine, quantitatively, if the infection progressed faster. Rather than ask the authors to conduct such experiments at this stage, I would suggest that they simply delete or modify this statement, and equally importantly, indicate within their paper that definitive proof of the role of these genes in immunity awaits infection bioassays. The omission of these experiments does not detract from their findings that these genes may play a role in development and immunity. Perhaps many of these genes play dual roles (e.g. Toll’s role in dorsal patterning during embryogenesis, and then in immunity later), being more critical for development during the larval phase, and then more relevant as immunity genes in adults.

2. Pages 19 notes mortality of the larvae after only three days of feeding on five (5) of the 11 dsRNAs [Toll5s, Serpin2, PGRP-LB, PGRP-LB2b, Chaoptin], and on page 20, it was noted that five dsRNAs induced significant RNAi-mediated knockdown by this time-point [PGRP-LB2b, Toll5b, Toll7b, Toll9b, I-type1]. However, the two lists do not really correspond, with the exception of PGRP-LB2b. Can the authors offer some explanation as to why mortality was observed in those 3-day feeders when transcript knockdown was not considered significant (e.g. PGRP-LB, Chaoptin), or was only mildly knocked down (e.g. ~10% for Toll5a, 20% Serpin2)? Can the authors relate the early mortality to those particular genes’ functions, and similarly, indicate why some of the other dsRNAs took considerably longer (15 days) to have an impact on the survivorship of the insects?

3. On page 21, the authors noted that PGRP-LB2a, when knocked down by RNAi, resulted in arrested development, but commented that PGRP-LB2b did not have a role in development, as no such phenotype was observed when the latter was knocked down. However, as noted above, PGRP-LB2b killed the insects early, so could that not also mean that the 2b gene is essential for development, and in fact, is more critical than 2a? Some further elaboration on the possible roles of 2b could be helpful.

4. The authors comment on page 19 that PBS treatments were used as negative controls, but there is no evidence of these treatments in any graphs.

5. On page 23, the authors state “Thus, LgI-type-1 exhibits isopeptidase activity…” but as there is no experimental data for this, it would be better to state “Thus, LgI-type-1 may have isopeptidase activity”.

Minor points (minor typos).
6. Page 9, Line 7. Change: “highly efficient of RNAi to” to “efficiency of RNAi in”
7. Page 13, line 6. Delete “of”
8. Page 13, line 8. Insert “two” before “pooled” for clarity – if I understood the method accurately(?).
9. Page 20, line 16. Insert “and” before “helps”

·

Basic reporting

The article provides sufficient background information to understand the relevance of the stated research objective. The study's findings are relevant to this objective.

Below, I provide extensive suggestions that I hope the authors will find helpful for improving the manuscript. These recommendations seek to insure that the text is uniformly clear, that appropriate references are consistently cited, and that all figures and tables are properly prepared.

Note: Page numbers below refer to the PDF supplied to this reviewer, with the manuscript itself beginning on PDF page 4.

INTRODUCTION
p. 9, lines 1-9: This paragraph would benefit from revision. Most notably, it’s odd that in the last sentence an older reference (Terenius et al., 2011) is cited as “recently increased” information to counter the conclusions of a newer reference (Shukla et al., 2016). In addition, this sentence is grammatically incorrect. To better captures the authors’ intent, I recommend removing the opening phrase to produce this revised sentence: “However, RNAi is particularly successful when targeting genes involved in immune responses (Terenius et al., 2011).” Furthermore, the flow of this paragraph could be enhanced by switching the fourth (“The effectiveness of…”) and fifth sentences (“To date, applying RNAi technology…”) and then replacing “To date” with “However” (and now removing “However” from the last sentence of the paragraph).
p. 9, lines 10-18: In the first sentence, a research article on planthopper genomics is cited to support a factual statement about D. melanogaster and B. mori innate immune system. A more relevant reference should be cited instead.
The second sentence of this paragraph is somewhat convoluted and hence difficult to decipher; dividing this sentence in two at the first comma would help. The authors should clarify that PGRPs can activate the two pathways or induce a proteolytic cascade. Finally, it’s unclear if the final phrase (“which are critical for defending…”) refers to antimicrobial peptides or the activated/induced responses.
p. 10, lines 1-6: To better connect the opening sentence to the preceding paragraphs, insert a transition phrase such as “To identify immune-related genes…” at the beginning.
METHODS
p. 11, lines 8-11: The references cited at the end of this paragraph are not relevant to the closing sentence. A more appropriate reference might be Altshculer et al. (1990), McGinnis & Madden (2004), or another BLAST program reference.
p. 11, line 13-17: The Tamura et al. (2011) reference cited in the previous paragraph is more relevant to the first sentence of this paragraph. Appropriate references for the web sites identified in this paragraph are available at those sites (e.g., Clustal Omega, PMID:21988835; SMART, doi:10.1093/nar/gkx922).
p. 12, line 5: Only two supplementary tables were included in the package to review. In this package, Table S2 is labeled “Primers used for synthesising dsRNA and analysing candidate genes.”
p. 12, line 15: The reference is erroneously duplicated.
p. 13, lines 6-7: Change “of per biological replicates” to “per biological replicate.”
RESULTS
p. 14, line 5: It would be helpful to explicitly label the three groups in Table S1.
p. 14, line 9: The comma should be removed.
p. 14, lines 10-12: This sentence seems out of place. Perhaps it would fit better at the start of the next subsection, “Immune signalling molecules” (p. 16, line 17).
p. 15, line 12: The authors should use the more specific term “ortholog” (instead of homologue).
p. 17, line 18: Fig. S2B displays an amino acid sequence alignment of the two L. glycinivorella Snake proteins, not of the three Serpin proteins, as this sentence would suggest.
p. 18, lines 5-7: Table S1 (“Details regarding the immune-related genes of Leguminivora glycinivorella”) includes two defensin genes, not just one.
p. 19, line 14: Not all cuticles are black in Fig. 5; the RNAi treatments causing black cuticles should be briefly identified in the text.
p. 19, line 16: This data is apparently not shown, so this should be (parenthetically) noted in the text.
DISCUSSION
p. 20, lines 15-19: This paragraph would benefit from revision. The first sentence has a grammatical error that should be corrected (and says insects three times). The second sentence should briefly note why these are of interest (or be removed). The third sentence should be clarified: Presumably hundreds of genes were identified in this transcriptome, with 41 encoding immune-related proteins. I suggest revising as follows: “In this study, we identified 41 genes in the L. glycinivorella transcriptome encoding components…”.
CONCLUSION
p. 23, lines 13-15: Two grammatical errors present: “would” should be “will” and the comma should be removed.
REFERENCES
Some references are not listed in alphabetical order, including Hughes (2012); Imler & Bulet (2005); Joga et al. (2016).
FIGURES
Figure 1: In the legend, “bootstrap values” should be changed to “bootstrap percentages” and D. melanogaster should be italicized.
Figure 2: (A) In the legend, “bootstrap values” should be changed to “bootstrap percentages.” (B) It should be noted in the legend that the blue bar indicates the transmembrane domain. A period is missing at the end of the last sentence of the legend.
Figure 3: <b>As noted elsewhere, the p-values must be corrected for testing multiple hypotheses; if p-values have been adjusted, this should be noted here (and the procedure used should be indicated in the Methods section). Also, the legend should mention what the error bars indicate (apparently not ±SD or ±SE since they are not symmetrical about the mean).</b> Finally, the legend should state that this is the “Mortality rate of larvae…”.
On the graph, the y-axis should be relabeled “Mean mortality (%)” and its maximum value should be 100%, since mortality cannot exceed this value. Also, the sample labels should be larger (and written at an angle to provide sufficient space).
Figure 4: Similar to Fig. 3, the legend should state that this is the average (or mean) body weight and should mention what the error bars indicate (apparently not ±SD or ±SE since they are not symmetrical about the mean), as well as if the p-values have been adjusted for testing multiple hypotheses. Unless the measurements were accurate to 0.01 mg, the numbers on the y-axis should be whole numbers.
<i>I’m surprised that no significant difference in mean body weight was detected for Lgitype-1, which shows similar means and variances as with LgToll-5b; in particular, 9d means and variances are less than for LgToll-5b.</i>
Figure 5: GFP control larvae should be included for comparison. What is CK in the 12th panel?
Figure 6: As noted for Figures 3 and 4, the legend should describe the error bars and whether p-values were adjusted for multiple testing.
SUPPLEMENTS
Fig. S1: If GNBP1 and 2 also from D. melanogaster, then this should be designated (as with DmGNBP3).
Table S1: The headers of Columns B and E point to footnotes that are not present at the bottom of the table. Abbreviations for some insect species noted in Columns J and K are included at the bottom of the table, but Bm and Pm are missing.
Table S2: The Column C header is misspelled (it should be “Product”).

Experimental design

The authors present a thorough examination of the potential of immune-related genes as RNAi targets for control of the soybean pod borer (SPB), Leguminivora glycinivorella. While the authors have conducted a largely well-designed experimental protocol to address their well defined research questions, the experimental description has a major flaw that must be addressed prior to publication. Two supplementary tables referred to in the text, which contain information necessary to replicate the experiments, are missing from the package provided to this reviewer. This major issue and a few more minor issues are detailed below.
METHODS
p. 11, line 4: The method used to create unigenes should be described or an appropriate reference should be cited.
p. 11, lines 8-11: <b>The list of immune-related genes used to complete the sequence similarity searches with tBLASTn should be included.</b> Table S1 (but not the cited Table S2) identifies some insect orthologs, but it’s not clear that this was the entire list used for the tBLASTn search. Perhaps this list is one of the missing supplementary tables (see “General comments for the author” section). This list should include GenBank identifiers (GI or Accession.Version numbers) and names of homologs from D. melanogaster or other organisms.
p. 12, line 5: Only two supplementary tables were included in the package to review. In this package, Table S2 is labeled “Primers used for synthesising dsRNA and analysing candidate genes.”

Validity of the findings

While the authors provide much convincing evidence in support of their conclusions, the manuscript has two major flaws that may compromise the validity of the findings. Foremost, the authors must explicitly address the issue of multiple hypothesis testing. Furthermore, features of the gene expression studies to confirm RNAi-mediated gene silencing must be remedied as described below. Finally, I also note some additional minor issues.
METHODS
p. 13, lines 4 & 17: <b>There is no indication how p-values were corrected for testing multiple hypotheses (i.e., Bonferroni or a more sophisticated correction). If the p-values have not been adjusted, then many of the study's conclusions are questionable.</b>
FIGURES
Figure 6: <b>How can GFP expression levels be used as a control? Unless larvae were transgenic, SBP does not express GFP. And what are expression levels relative to? I would expect expression levels to be relative to the 0d values for each gene, which should then all be set to 1.00, but this is not the case in Figure 6. Along with the description of the normalization method, the calculation of relative expression levels should also be explicitly described in the Methods section. </b>
RESULTS
p. 16, lines 11-13: <b>Several of these designations are problematic, since they incorrectly suggest close evolutionary relationships between distant members.</b> For example, the designation for LgToll-6c suggests that it is monophyletic with LgToll-6a and -6b. However, as shown in Figure 2, LgToll-6c shares the same common ancestor with LgToll-7a/7b and LgToll-6a/6b, and is therefore equally related to all four members. Similarly, the LgToll-5a/5b designations are misleading since these L. glycinivorella homologs are equally related to DmToll3, 4, and 5. Finally, LgToll-9b is also not monophyletic with LgToll-9a and should be renamed.
p. 19, line 6: Too many significant figures (42.86%) exaggerate measurement accuracy; change to 43-92%. Also refer to Fig. 3 here.
p. 19, line 7: The PBS data should be added to Figure 3. If not appropriate, then the text should mention (parenthetically) that this data is not shown.
DISCUSSION
p. 21, lines 7-10: It should be noted that the LgPGRPs presumably exhibit diverse activities, since this wasn’t tested directly.
p. 22, line 16: Earlier (p. 19, line 6) the maximum mortality rate was stated as 92%, not 93%. Correct either sentence to resolve this difference.
p. 23, lines 6-8: This sentence is incomplete. Consequently, I’m unable to understand the sentence that follows, which argues that LgI-type-1 exhibits isopeptidase activity.

Additional comments

The authors present a thorough examination of the potential of immune-related genes as RNAi targets for control of the soybean pod borer (SPB), Leguminivora glycinivorella. While the authors have conducted a largely well-designed experimental protocol to address their research questions and provide convincing evidence to support their conclusions, the manuscript has several major and numerous minor issues that should be addressed prior to publication. Foremost, two supplementary tables referred to in the text that are required to replicate the experiment are missing from the package provided to this reviewer. Second, the authors must explicitly address the issue of multiple hypothesis testing and some aspects of data presentation. Third, flaws in the gene expression studies to confirm RNAi-mediated gene silencing must be addressed. In addition to these significant issues (highlighted in <b>bold</b>), I also provide extensive suggestions that I hope the authors will find helpful for improving the manuscript.

Notes:
Page numbers below refer to the PDF supplied to this reviewer, with the manuscript itself beginning on PDF page 4.
As mentioned, the manuscript refers to Tables S1-S4, but the downloaded package contained only two tables. When the text refers to Tables S2 and S3, it apparently corresponds to Tables S1 (“Details regarding the immune-related genes of Leguminivora glycinivorella”) and S2 (“Primers used for synthesising dsRNA and analysing candidate genes”), respectively, in the downloaded package.

·

Basic reporting

manuscript has clear and concise English use and sentence structure. Sufficient references in the field of study were cited. The article, figures, and tables are suitable for professional publication.
RAW DATA SHARING" Sequences mRNA are listed in file, which is good, IF these have been submitted into NCBI, or similar public database the assembled sequences accession numbers and the website where located NEED to be included. Authors have tried to submit short archive sequences, which is not sufficient. Please add into text where suggested (Results).
Conclusions and summary support hypothesis.

Experimental design

Manuscript fits within the aims and scope of journal.

Research is of significant interest and active research field, filling and adding to the knowledge gap in RNAi.

Investigation appeared to be of suitable replication and analyses, with suitable use of statistics and gene analysis software, and standards.

Methods described in sufficient detail and information to replicate.

Validity of the findings

Impact and potential assessment of application was reasonable and followed from the results logically.

Conclusions well stated, and linked to results, displayed well in the figs/tables included.

Data files with mRNA and protein sequences, phylogeny, etc. well presented and support conclusions.

Additional comments

Well done, nice research and of signficant impact for lepidopteran studies. It was interesting that with so many lepidopteran species cannot induce RNAi post feeding, yet your insect appeared to work just fine.
A future study may want to try and determine WHY the RNAi method worked so well in your experiment.

---

## Round 0.2 · Minor Revisions

Thank you for your work to revise your manuscript and address comments from the two reviewers. Both reviewers agree that the manuscript is improved. I have a few remaining issues that should be addressed before your manuscript can be accepted.

Thank you for your edits to the transcriptomics section of the Methods. I was confused about one detail. Were the six pooled larvae randomly collected from the three biological replicates of 50 first-instar larvae? Can you clarify that part of the method. Also, unless I am missing the details, your methods section still does not list the kit used for library construction. Can you please add this detail?

Please check that supplemental figure 1 is in the correct file format, as reviewer 2 was not able to open the file. The .emf file format may be Windows specific. If so, please convert this file to a format that can be opened by other operating systems.

Thank you for your additional information about the qRT-PCR experiments. Your added reference (Meng et al. 2017b) does not seem to provide data showing that beta-actin is a verified reference control gene for SBP. The added citation does show that this gene has been previously used as a reference control, but does not provide validation that it is expressed equally across sample types or is unaffected by RNAi treatment. Do you know of any literature that verifies the use of this reference gene? I think it would also still be helpful to detail the primer and sample concentrations used in the reactions, the types of negative controls used (non-template, -RT?) and to provide the efficiencies for the primer pairs used.

Please also review your text for grammatical errors. Reviewer 2 commented on some general editing that needs to be done across the manuscript. I have noted several needed edits below, which are mostly associated with portions of the manuscript that have been edited:

Line 83 – “microinjection” should be lower case
Line 87 – “While applying” should not have a comma
Line 168 – “They were then used as . . .” – remove “being”
Line 184-185 – the last line of this paragraph needs to be corrected.
Line 190 – P-values were “corrected” by . . .
Line 274 – Remove the additional “e”
Line 311 – “In addition to sequence differences . . .”
Line 326 – Remove extra “f”
Line 359 – “SPB immune”
Line 447 – The Meng et al. reference should be “2017b”

Reviewer 1 ·

Basic reporting

The authors have made considerable revisions to the manuscript and I am satisfied with the changes.

Experimental design

The Methods have had sufficient revisions and are now clear.

Validity of the findings

The data are robust and the conclusions are appropriate.

Additional comments

I'm satisfied with the revisions.

·

Basic reporting

The author has addressed my comments and recommendations from my previous review. Some of the changes in the text have grammatical errors, but can be comprehended.
The figures have been revised as I recommended. However, I was unable to open the .emf file of Fig. S1 on my Macintosh computer.

Experimental design

The author has addressed my comments and recommendations from my previous review.

Validity of the findings

The author has addressed my concerns regarding Bonferonni correction due to testing multiple hypotheses.

Additional comments

The author has done a thorough job addressing my concerns and suggested revisions. Good work!

---

## Round 0.3 · accepted · Accept

Thank you for these final edits and for all of your work to improve this manuscript. I am happy to now accept your manuscript for publication in PeerJ. Congratulations on this important contribution to the use of RNAi to control agricultural pests.

I did notice a few edits that could be made when you review your proofs:

Line 83: should read “microinjection of RNAi”
Line 87: Add a space between “While applying”
Line 170: Remove the extra space between “then used”

You will be given the option to make the reviews of your manuscript available to readers. Please consider doing so as this review record can be a great resource for readers of your paper and contributes to more transparent science.

Thank you again for your contribution.

#